# The Role of *PnTCP2* in the Lobed Leaf Formation of *Phoebe neurantha* var. *lobophylla*

**DOI:** 10.3390/ijms232113296

**Published:** 2022-10-31

**Authors:** Bing Sun, Xinru He, Fengying Long, Cui Yu, Yongjun Fei

**Affiliations:** 1College of Horticulture and Gardening, Yangtze University, Jingzhou 434025, China; 2Academy of Agricultural Sciences, Industrial Crops Institute of Hubei, Wuhan 430064, China

**Keywords:** *P. neurantha* (Hemsl.) Gamble var. *lobophylla*, transcriptome sequencing analysis, lobed leaves, miRNA319a, *PnTCP2*

## Abstract

A lobed leaf is a common trait in plants, but it is very rare in Lauraceae plants, including species of *Phoebe.* In the study of germplasm resources of *Phoebe neurantha*, we found lobed leaf variant seedlings, and the variation could be inherited stably. Studying the lobed leaf mechanism of *P. neurantha* var. *lobophylla* can offer insight into the leaf development mechanism of woody plants. RNA-seq and small RNA-seq analysis results showed that a total of 8091 differentially expressed genes (DEGs) and 16 differentially expressed miRNAs were identified in *P. neurantha* var. *lobophylla*. Considering previous research results, a leaf margin morphological development related miRNA, pne-miRNA319a, was primary identified as a candidate miRNA. Target gene prediction showed that a total of 2070 genes were predicted to be the target genes of differentially expressed miRNAs. GO enrichment analysis of differentially expressed target genes suggested that *PnTCP2* is related to lobed leaf formation. The TRV-VIGS gene silencing of *PnTCP2* led to lobed leaves in *P. neurantha* seedlings. The downregulation of *PnTCP2* led to lobed leaves. The yeast two-hybrid test and bimolecular fluorescence complementation test confirmed that the PnTCP2 protein interacted with the PnLBD41 protein. Based on the expression analysis of gene-silenced leaves and RNA-seq and small RNA-seq analysis results, pne- miRNA319a and *PnLBD41* might also play important roles in this process. In conclusion, *PnTCP2* plays an important and vital role in the formation of the lobed leaves of *P. neurantha* var. *lobophylla.*

## 1. Introduction

*Phoebe neurantha* is an evergreen tree of the Lauraceae family. *P. neurantha* (Hemsl.) Gamble var. *lobophylla* Y. J. Fei (Figure 1A) is a new variety with a different leaf morphology that was found in a study of *P. neurantha* germplasm resources [1]. Compared with the leaves of *P. neurantha* (PnZC) (Figure 1B), the leaves of *P. neurantha* var. *lobophylla* (PnBY) are lobed (Figure 1D). According to the Flora of China records (FOC Vol. 7 @ efloras.org), *Phoebe* Nees has entire leaves, and there are few lobed leaves in the Lauraceae family. An exception to this is *Sassafras* plants, which have whole- or shallow-lobed leaves. Therefore, the emergence of the variation in the leaf margin of *P. neurantha* var. *lobophylla* has aroused great interest. The leaf-lobed trait not only has a significant impact on the photosynthetic efficiency of the plants and their adaptability to the environment but also enhances their ornamental value as garden plants.

Leaf margin development is regulated by many genes. Researchers have identified many genes related to leaf margin morphology and development in different plants, such as *AtCUC2* [2] and *AtTCP4* [3] in *Arabidopsis*, *CcLBD41* [4] in *Celosia cristata*, *MtLMI1* [5] in *Medicago truncatula*, and *KNOX1*, which is more widely reported (in addition to reports on *Arabidopsis* [6], there are also reports on *Solanum lycopersicum* [7], fern *Mickelia scandens* [8], *Pisum sativum* [9], *Medicago truncatula* [5], and other plants). The leaf development process of plants is unique. Leaves have polymorphisms and complexities that are regulated by environmental factors. However, plant leaf development also has a basic pattern, with stability and order. Many studies have shown that leaf margin development is affected by environmental and genetic factors, such as light, moisture, hormones, temperature, transcription factors, and microRNAs (miRNAs) [5,10,11]. miRNAs are endogenous non-coding RNAs. These RNAs widely exist in the genomes of animals, plants, and microorganisms. They play regulatory roles in the expression of genes related to growth, development, and stress resistance. Moreover, miRNAs are involved in the growth and development of plants, such as in regulating leaf shape [12]. From the perspective of plant evolution and phylogeny, the formation of plant leaf morphology is mainly determined by genes, miRNAs, and other regulatory factors. A number of miRNAs and their target genes have been shown to determine leaf development, such as miRNA156-*SPL* (*SQUAMOSA PROMOTER BINDING PROTEIN-LIKE*) [13], miRNA164-*CUC* (*CUP-SHAPED COTYLEDON*) [14], and miRNA319-*TCP* (*TEOSINTE BRANCHED 1/CYCLOIDEA/PCF*) [15,16].

Many genes and miRNAs that are involved in the regulation of leaf development have been cloned and identified with the help of *Arabidopsis thaliana* [17] and other model plant mutants [18,19]. These results provide a good basis for studying the mechanism of lobed leaf formation. However, due to the slow growth of woody plants and the low frequency of leaf margin variation under natural conditions, research on woody plant leaves is limited [20]. This research will lay a foundation for revealing the regulation network of lobed leaf formation in *P. neurantha* var. *lobophylla.*

## 2. Results

### 2.1. Identification and Analysis of DEGs

The transcript sequence obtained by Trinity splicing was used as the reference sequence for subsequent analysis. The longest cluster sequence obtained from Corset hierarchical clustering was used as the unigenes for subsequent analysis. The lengths of the transcripts and unigenes were determined (Appendix A). After Corset hierarchical clustering, the N50/N90 of the sequence increased from 1170/319 bp of the transcripts to 1451/497 bp of unigenes, and the mean length of the sequence also increased from 764 to 1043 bp. The results showed that unigenes could better represent the transcripts for subsequent analysis. The differential expression analysis showed that 169,812 and 152,100 unigenes were expressed in PnZC and PnBY, respectively, of which 122,447 unigenes were expressed in both, with only 47,365 unigenes in PnZC and 29,653 unigenes in PnBY (Figure 1E). A total of 8091 DEGs (Appendix A) were identified in PnBY with the help of DESeq, of which 4129 DEGs were upregulated and 3962 DEGs were downregulated in PnBY (Figure 1G).

### 2.2. Identification and Expression Patterns of Known miRNAs

A total of 190 mapped mature and 198 mapped hairpin RNAs were identified in PnBY and PnZC (Figure 1F). Among these, 22 known miRNAs were identified. However, pne-miRNA482a was only identified in PnBY, and three known miRNAs (pne-miRNA160d, pne-miRNA169q, and pne-miRNA397) were expressed only in PnZC. Only two known miRNAs (pne-miRNA162a and pne-miRNA319a) were found to be differentially expressed in PnBY (Figure 1H).

### 2.3. qRT-PCR Validation of Related Genes and miRNA

To validate the transcriptome sequencing data, *PnKNAT6*, *PnLBD41*, *PnTCP2,* and pne-miRNA319a were chosen for validation of expression by qRT-PCR analysis (Figure 1I). In the qRT-PCR analysis, the expression patterns of these genes and pne-miRNA319a were approximately consistent with the results of transcriptome sequencing. These results indicate that the transcriptome sequencing data were reliable.

### 2.4. Prediction and Enrichment Analysis of the Target Genes of Differentially Expressed miRNAs

A total of 2070 target genes (Appendix A) of differentially expressed miRNAs were predicted in all unigenes. Go enrichment analysis (Appendix A) showed that the target genes were significantly enriched under the classes “Molecular function” and “Cellular component”. The GO subcategories (Figure 2A) related to morphogenesis were listed, which showed that the “leaf morphology” subcategory was significantly enriched. A total of 19 target genes (Appendix A) were enriched in the “leaf morphology” subcategory. Coincidentally, 12 of them are target genes of pne-miRNA319a. Cluster thermogram analysis (Figure 2C) showed that *PnTCP2* (Cluster-15957.72980) was downregulated in the PnBY samples, which was consistent with the targeting characteristics of pne-miRNA319a. Based on the target gene prediction results, *PnTCP2* (Cluster-15957.72980) is a potential target gene of pne-miRNA319a. Furthermore, KEGG enrichment analysis (Figure 2B) showed that 23 KEGG pathways such as “Amino acid metabolism”, “Ubiquinone and other terpenoid-quinone biosynthesis”, and “Ether lipid metabolism” were significantly enriched (corrected *p*-value < 0.05). Studies have shown that the overexpression of miRNA319 promotes the formation of leaf serrations, while the inhibition of endogenous miRNA319 activity results in the opposite phenotype [21,22]. In conclusion, pne-miRNA319a-*PnTCP2* was the candidate module related to lobed leaf formation.

### 2.5. Gene Cloning and Gene Structure Analysis

PCR amplification was performed with specific primers (Appendix A). Subsequently, the amplified products were detected by 1% agarose gel electrophoresis (Figure 3A). The results showed that the PCR products of *PnTCP2* and *PnLBD41* were about 1400 and 900 bp, respectively. The PCR products were sequenced, and the results were consistent with the transcriptome data. The conserved domains of the PnTCP2 and PnLBD41 proteins were analyzed by smart tool (https://smart.embl-heidelberg.de/ (accessed on 6 December 2021)). The analysis results showed that the PnTCP2 protein (Figure 3C) had the TCP domain. The PnLBD41 protein (Figure 3C) had the LOB domain. Protein conservative domain analysis showed that PnTCP2 was a TCP family transcription factor and that PnLBD41 was a LOB family transcription factor. At the same time, the mRNA site of *PnTCP2* targeted by pne-miRNA319a was predicted and analyzed using the psRNATarget online tool (Figure 3D). The mRNA cleavage site of pne-miRNA319a in *PnTCP2* was located in the open reading frame (ORF) region between 1102 and 1121 bp, and the expectation value was 3, indicating that the possible target gene of pne-miRNA319a was *PnTCP2*.

### 2.6. Functional Analysis of PnTCP2

SiRNA was predicted using sidirect v.2.0 (http://sidirect2.rnai.jp/ (accessed on 16 January 2020)). According to the location of the siRNA, a 332 bp (Figure 4A) fragment was selected as a gene silencing fragment *(VTCP2*). The fragment was amplified by PCR using specific primers (Appendix A). Electrophoresis and positive validation results showed that the fragment size was in line with expectations (Figure 4B). The identification of positive plants (Figure 4C) showed that a total of four pTRV2-*VTCP2* seedlings with lobed margins were identified (Figure 4D). After gene silencing, the expression of *PnTCP2* was significantly downregulated (Tukey, *p* < 0.05), indicating that *PnTCP2* was silenced in leaves (Figure 4E). The leaves of pTRV2-*VTCP2* seedlings (Figure 4(Da–Dd)) had obvious lobes compared with the control treatment (Figure 4(De)). Therefore, it was not difficult to conclude that silencing *PnTCP2* led to the growth of lobed leaves in the seedlings of *P. neurantha*. At the same time, the expression level of *PnLBD41* was significantly downregulated (Tukey, *p* < 0.05) in pTRV2-*VTCP2* seedlings. By contrast, the expression level of *PnKNAT6* was significantly upregulated (Tukey, *p* < 0.05).

To confirm the possible protein interaction, the interaction of PnTCP2 with PnLBD41 was verified by bimolecular fluorescence complementation (BiFC) (Figure 5A). The results showed that PnTCP2 interacted with PnLBD41. In further research, yeast two-hybrid validation was used to support the results of BiFC verification, and the protein interaction of PnTCP2–PnLBD41 was confirmed (Figure 5B).

## 3. Discussion

Plants have great differences in leaf morphology, which is mainly due to the morphological and structural characteristics of their leaf margins [23]. Leaf shape development is a complex process that involves the interaction among miRNAs, genes, and hormones for multi-layer network regulation.

Transcriptome sequencing analysis has become an important method in the study of plant morphological development. Similarly, it has also been reported for the morphological development of plant leaves. A total of 8767 unigenes were upregulated and 8379 unigenes were downregulated in the comparison of two *Betula pendula* plants with different leaf shapes at the transcriptome level [24]. Researchers have identified 43 differentially expressed genes (DEGs) that potentially regulate leaf shape in *Perilla frutescens* using transcriptome sequencing and co-expression analysis, which may potentially regulate leaf shape [25]. In this study, the Corset hierarchical clustering method was used to select unigenes, which not only achieved the goal of aggregating redundant transcripts, but also improved the detection rate of differentially expressed genes [26,27,28]. In this study, RNA-seq and small RNA-seq were performed on the leaf buds of *P. neurantha* (entire leaf) and *P. neurantha* var. *lobophylla* (lobed leaf) to mine the genes and miRNAs that regulate the occurrence of lobed leaves. A total of 8091 DEGs were identified, of which 4129 were upregulated and 3962 were downregulated (Figure 1G). Furthermore, 16 differentially expressed miRNAs were found in PnBY, comprising 11 downregulated and five upregulated miRNAs (Figure 1H). Among these, the known differentially expressed miRNAs were pne-miRNA319a and pne-miRNA162a, which were upregulated in PnBY.

In *Arabidopsis*, miRNA319, also known as miRJAW, affects leaf complexity by targeting *TCPs* (*TCP2*, *TCP3*, *TCP4*, *TCP10,* and *TCP24*) [29,30,31]. Previous studies have shown that the overexpression of miRNA319 promotes the formation of leaf serration, while the inhibition of miRNA319 activity shows the opposite phenotype [21]. Through the silencing of *TCP2* in the leaves, the leaf margin showed a strong lobed phenotype [32]. The overexpression of the *ApTCP2* and *ApmTCP2* genes (synonymous change in the *ApTCP2* sequence to reduce the cleavage of miRNA319) of *Acer palmatum* in the *jaw-D* mutant of *Arabidopsis* reduced or eliminated the lobes on the leaf margin [16]. Thus, the downregulated expression of *TCP2* was conducive to the development of lobed leaves [33], resulting in lobed leaf margins in *P. neurantha* var. *lobophylla*. In *Arabidopsis*, *AtTCP2* is one of the target genes of miRNA319 [15]. The significantly upregulated expression of miRNA319 resulted in the inhibition of the expression of the AtTCP2 transcription factor and the lobes of the leaf margin [22]. In this study, the target genes of differentially expressed miRNAs were predicted in unigenes, and 2070 target genes were predicted. GO enrichment analysis showed that target genes were significantly enriched in the “leaf morphology” subcategory (Figure 2A), with a total of 19 target genes (Appendix A) enriched in this subcategory. Coincidentally, 12 (all TCP family genes) are target genes of pne-miRNA319a, especially *PnTCP2* (Cluster-15957.72980), whose expression pattern was opposite to that of pne-miRNA319a. In addition, this study conducted gene silencing on *PnTCP2* and found four pTRV2-*VTCP2*-positive seedlings with lobed leaves (Figure 4D). The silencing of *PnTCP2* promoted lobes on the leaf margins of the *P. neurantha* seedlings. VIGS technology has the advantages of independent genetic transformation, convenient inoculation, and high silencing efficiency, which can achieve the silencing of targeted genes in contemporary plants [34]. However, there are still some limitations, such as the inability to completely silence genes [35]. In this study, we successfully silenced *PnTCP2* in the leaves of *P. neurantha* seedlings using VIGS technology and obtained the lobed leaf phenotype. However, the lobes of the gene-silenced leaves were very tiny compared with the lobed leaves of *P. neurantha* var. *lobophylla*. The reason for this phenomenon may be that the mRNA of *PnTCP2* was not completely silenced, and the surviving mRNA weakened the degree of the phenotype. Moreover, the regulation of plant leaf margin morphology involves the synergistic effects of multiple genes (including *TCP4* [3], *CUC2* [36], and *LMI1* [5]). Genes such as *PnTCP4* were not silenced. The expression of these genes weakens the phenotypic changes caused by *PnTCP2* silencing. However, the fact that the silencing of *PnTCP2* caused the leaves of *P. neurantha* seedlings to be lobed cannot be concealed. Therefore, *PnTCP2* and the possible pne-miRNA319a-*PnTCP2* module play an important regulatory role in the lobed leaf formation of *P. neurantha* var. *lobophylla.*

In conclusion, due to the targeted inhibition of miRNA319 on *TCP2* [16,31], the transcriptional expression of *PnTCP2* was inhibited by the upregulated expression of pne-miRNA319. In *Arabidopsis*, the mutation of *Atas2* or the overexpression of miRNA319 led to lobed leaf formation. Moreover, the overexpression of miRNA319 in *Arabidopsis* plants with the *Atas2* mutation deepened the degree of the leaf lobe. In addition, AtAS2 inhibits the expression of AtKNAT6 by binding to the promoter [37]. Surprisingly, the AtAS2-AtTCP2 dimer also has similar functions [37]. In *Arabidopsis*, AtLBD41 is an ASL38 protein that is a member of the AS2/LOB protein family [38], indicating that LBD41 and AS2 have similar functions. Coincidentally, LBD41 has also been reported to inhibit the expression of KNAT6 in leaves [37]. However, the overexpression of *AtKNAT6* in *Arabidopsis* could enhance the complexity of the leaf margin [39].

In this study, the PnTCP2 protein interacted with the PnLBD41 protein (Figure 5A,B), indicating that it also inhibited the expression of *PnKNAT6.* The expression levels of *PnTCP2* and *PnLBD41* were significantly downregulated in the leaf buds of both *P. neurantha* var. *lobophylla* and pTRV2-*VTCP2* seedlings (Figure 4E)*,* resulting in the upregulation of *PnKNAT6* expression, which was conducive to lobed leaf formation.

## 4. Materials and Methods

### 4.1. Plant Materials

Samples were taken from four-year-old healthy *P. neurantha* (PnZC) and *P. neurantha* var. *lobophylla* (PnBY) plants. Both *P. neurantha* and *P. neurantha* var. *lobophylla* were transplanted to the germplasm resources nursery in 2016. The germplasm resources nursery is located in the Agricultural Science and Technology Industrial Park of Yangtze University, Jingzhou National Hi-Tech Zone, Hubei, China. The seeds were collected from the subtropical evergreen–deciduous broad-leaved mixed forest in the Tujia and Miao Autonomous Prefecture of Enshi, Hubei Province (E 108°23′12″–110°38′08″, N 29°07′10″–31°24′13″), China. The top buds of lateral branches (not yet germinated) of healthy plants (PnZC and PnBY) were randomly collected as experimental materials. The buds of *P. neurantha* were randomly divided into two groups (three replicates), as were the buds of *P. neurantha* var. *lobophylla*. After the shoots were collected from the branches, the scales were removed and the buds were immediately placed in liquid nitrogen. Finally, the buds were stored at −80 °C in preparation for the RNA extraction. The RNA was isolated from each sample and was used for RNA-seq and small RNA-seq analyses.

### 4.2. Construction of Small RNA Libraries and Deep Sequencing

The total RNA was extracted using TRIzol reagent (Life Technologies, Beijing, China) according to the manufacturer’s protocol. For the RNA-seq, the RNA was enriched by magnetic beads with Oligo (dT) after the samples were qualified. The purified double-stranded cDNA was end-repaired, A-tailed, and ligated to a sequencing linker, and AMPure XP beads were used for fragment size selection. PCR amplification was performed, and the PCR product was purified using AMPure XP beads to obtain the final library. After the insert size met expectations, the effective concentration of the library was quantified accurately by Q-PCR (library effective concentration > 2 nM) to ensure the quality of the library.

For the small RNA-seq, a small RNA library was established and sequenced with 3 µg total RNA in each sample. NEBNext^®^ Multiplex Small RNA Library Prep Set for Illumina^®^ (NEB, Ipswich, MA, USA) was used to construct the small RNA libraries following the manufacturer’s recommendations. The index codes were then added to the attribute sequence for each library. Through TruSeq SR Cluster Kit v3-cBot-HS (Illumina, San Diego, CA, USA), the index-coded samples in the cBot Cluster Generation System were clustered according to the manufacturer’s instructions. Subsequently, the library of RNA-seq and small RNA-seq was sequenced using an Illumina Hiseq 2500/2000 platform to generate the required reads. The raw data obtained from sequencing have been submitted to the NCBI database. BioProject number in the NCBI database is PRJNA593676 (See Appendix A for details).

### 4.3. Analysis of Sequencing Data

Trimmatic v0.38 [40] was used for the filtering and quality control of the raw data. The quality reads were cleaned using Trinity v2.4.0 [41] for de novo assembly to obtain the transcripts. The transcripts were hierarchically clustered using Corset v1.05 [26] to obtain the longest cluster sequence as a unigene sequence for subsequent analysis.

For the small RNA-seq, the raw reads in fastq format were first processed. During this step, clean reads were obtained by deleting reads that contained poly-N, poly-A, T, G or C, 5′ adapter contaminants, no 3′ adapter or insert tags, and low-quality reads from the raw reads [42]. At the same time, the Q20, Q30, and the content of GC in the raw reads were calculated [42]. Then, the clean reads with a length of 18–25 nucleotides were used for downstream analysis. In order to analyze their expression and distribution in reference sequences, Bowtie [43] software was used to accurately map the small RNA tags to reference sequences (the transcripts were spliced in RNA-seq with Trinity v2.4.0 software).

### 4.4. Analysis of DEGs and Differentially Expressed miRNAs

RSEM v1.2.15 [44] software was used to calculate the expression level by mapping the clean reads of each sample to the transcripts obtained by Trinity splicing. The differential expression analysis of the DEGs and differentially expressed miRNAs from the two groups (*P. neurantha* var. *lobophylla* vs *P. neurantha*) was conducted using the DESeq [45] R package (1.10.1). DEGs were recognized if the fold-change ≥ 2.00 and false discovery rate (FDR) < 0.05 were met simultaneously. A corrected p-value less than 0.05 was considered to be significantly different by default. The expression of known miRNAs in each sample was determined and normalized using the Transcripts Per Kilobase Million (TPM) [46] method. Formula: normalized expression level = (read count × 1,000,000)/libsize (libsize is the sum of the sample miRNA read counts).

### 4.5. Known miRNA Alignment, Target Prediction, and GO and KEGG Pathway Analyses

The mapped small RNA tags were used to search for known miRNAs in the miRNA database, miRNABase 20.0 [47]. With the use of miRNABase 20.0 as the reference database, modified software miRNADeep2 [48] and srna-tools-cl were used to predict the pre-miRNAs and draw the secondary structures [49].

In the aspect of target gene prediction, psRNATarget [50] was used to predict the target genes of miRNA. Due to the unavailability of genomic sequence data, the unigenes of RNA-seq were used for the prediction of putative miRNA targets. Based on the annotation results of eggNOG-mapper v2 (http://eggnog-mapper.embl.de/ (accessed on 2 June 2022)), TBtools [51] software was used for the GO and KEGG enrichment analyses of target genes.

### 4.6. Quantitative Real-Time PCR Analysis

The Quantitative Real-time PCR (qRT-PCR) analysis of transcriptome data was performed with the same samples used in transcriptome sequencing. The total RNA was extracted using the EASY spin Plus Plant RNA Kit (Aidlab, Beijing, China) according to the manufacturer’s protocol. The miRNAcute Plus miRNA First-Strand cDNA Synthesis Kit (TIAN GEN, Beijing, China) uses the method of adding poly-A to carry out the reverse transcription of the miRNA first-strand cDNA. The miRNAcute Plus miRNA qPCR Kit (TIAN GEN, Beijing, China) was then used to quantify the miRNA transcript level. The qPCR was performed using a Line-Gene 9600 Plus Real-Time PCR detection system (Bio-Rad, Hercules, CA, USA) with SYBR^®^ Green Real-time PCR Master Mix (Takara, Dalian, China) according to the manufacturer’s protocols. All measurements contained a negative control with sterile water (no cDNA template). With the use the National Center for Biotechnology Information (NCBI) primer designing tools to design qPCR primers for the target genes, *EF-1α* and the *U6* gene (for the miRNAs) were selected as the internal reference genes. All primers used in this research are listed (Appendix A). All reactions were performed in triplicate. The data were analyzed according to the comparative CT method (2^−ΔΔCT^) [52].

### 4.7. Protein Sequence Analysis

The analyzed sequences comprised 23 sequences selected according to the results of a GenBank BLAST search (https://blast.ncbi.nlm.nih.gov/Blast.cgi (accessed on 10 December 2021)), as well as the protein sequence of PnTCP2. Sequences were aligned using MEGA7.0. The aligned dataset was analyzed by the Maximum Likelihood (ML), using a rapid bootstrapping algorithm and 1000 replicates, followed by an ML tree search. SMART Tools (https://smart.embl-heidelberg.de/ (accessed on 11 December 2021)) software was used to predict the conserved gene domains.

### 4.8. Vector Construction

The correctly sequenced fragments were constructed into pGBKT7-Bait (pGBKT7-PnTCP2 and pGBKT7-PnLBD41), pGATD7-prey (pGATD7-PnAS1 and pGATD7-PnLBD41), and pTVR2-VTCP2 vectors using the Gateway™ BP Clonase™ II Enzyme mix (Invitrogen, Carlsbad, CA, USA) Kit and Gateway™ LR Clonase™ II Enzyme mix (Invitrogen, Carlsbad, CA, USA) Kit according to the instructions provided by the manufacturer. With the use of the CV19 One-step Seamless Cloning Kit (Aidlab, Beijing, China), the coding sequence (CDS) (without the stop codon) of *PnLBD41* was cloned into the polyclonal site of the p1300-SPYNE(R) vector to obtain the CFPN-*PnLBD41* fusion vector. The *PnAS1*-CFPC and *PnTCP2*-CFPC fusion vectors were constructed in the same way.

### 4.9. Protein Interaction Verification

The Yeastmaker™ Yeast Transformation System 2 Kit (Clontech, Daliang, China) transformed the extracted and purified plasmid into yeast strains Y2H and Y187. The Y2H strain containing pGBKT7-bait and the Y187 strain containing pGATD7-prey were matched on a small scale. After mating, the mating products were screened on quadruple dropout media (SD/–Leu–Trp–His–Ade/X-α-Gal) added to 100 ng/L Aureobasidin A (AbA). The experimental process refers to the instructions provided by the manufacturer of the Matchmaker Gold Yeast Two-Hybrid System (Clontech, Daliang, China).

After purification, the vector was transformed into the *Agrobacterium* EHA105 strain for the transient expression of tobacco (*Nicotiana benthamiana*) leaves [53]. Tobacco was incubated in the dark for 24 h and then returned to cultivation under light with normal water management after tobacco injection. The fluorescent signals of mesophyll cells were detected and photographed under a Leica TCS-SP8 SR microscope (Leica Microsystems) 3–5 days after inoculation.

### 4.10. TRV-VIGS Gene Silencing

The pTVR2-VTCP2, pTVR2-gateway, and pTVR1 vectors were transformed into the *Agrobacterium* EHA105 strain. Bacterial liquid PCR ensured that the plasmids were successfully transferred. Subsequently, the bacterial solution was transferred to sterile 250 mL glass flasks containing 100 mL of liquid LB medium (containing 0.15 mmol/L acetosyringone (AS)). The bacterial solution was cultured to OD_600_ = 1.0 at 200 rpm in the dark at 28 °C. The culture was centrifuged at 2000 rpm for 10 min. The supernatant was discarded, and the bacteria were washed three times with sterile liquid LB medium. The bacteria were resuspended to OD_600_ = 1.0 with infection buffer (containing 50 mmol/L 2-(N-morpholino) ethanesulfonic acid (MES), 0.15 mmol/L AS, 10 mmol/L MgCl_2_, 0.02% Silwet L-77, and 5 mL/L Tween-20), and pTRV1 and pTRV2 (including pTVR2-gateway, the control treatment) bacterial solutions were mixed in equal volume to prepare the infection solution. Then, the infection solution was resuscitated and cultured at 200 rpm in the dark at 28 °C for 30 min. Thirty seeds of *P. neurantha* that were pretreated (small wounds on the cotyledons of the seeds were carefully cut with a clean dissecting knife to facilitate the infection of *Agrobacterium*) were selected and placed into a 250 mL triangular flask. The infection liquid was added to completely immerse the seeds in the liquid. The triangular flask was placed in a vacuum kettle under low pressure at room temperature for 10 min, and the air pressure was restored for 10 min. The same operation was repeated three times. The treated seeds were transferred to a 500 mL beaker padded with 4–5 layers of wet filter paper and cultured under a day/night regime of 23/17 °C and 12 h light. The filter paper was kept moist during the culture. After the seed leaves grew, the morphological changes in the leaves were observed by comparing them with those of the control treatment. The gene silencing efficiency was determined using qRT-PCR analysis.

## 5. Conclusions

In this study, RNA-seq and small RNA-seq analyses revealed that the differentially expressed pne-miRNA319a was related to the formation of leaf margin lobes in *P. neurantha* var. *lobophylla*. Subsequently, GO enrichment analysis of the target genes of differentially expressed miRNAs showed that the possible pne-miRNA319a-*PnTCP2* module played an important role in the formation of leaf margin lobes in *P. neurantha* var. *lobophylla*. Lobed leaves were found in seedlings of *P. neurantha,* which successfully silenced *PnTCP2*, indicating that the downregulation of *PnTCP2* promoted the formation of lobed leaves. The qRT-PCR analysis showed that the silencing of *PnTCP2* resulted in the downregulation of *PnLBD41* in the pTRV2-*VTCP2* seedlings. By contrast, the silencing of *PnTCP2* upregulated the expression of *PnKNAT6* in the pTRV2-*VTCP2* seedlings. Fortunately, the reliability of the PnTCP2-PnLBD41 protein interaction was verified by yeast two-hybrid and BiFC experiments. In conclusion, combined with previous studies, the possible regulatory networks (Figure 6) can be summarized: The upregulated expression of pne-miRNA319a in leaf buds inhibited the expression of the target gene *PnTCP2*, resulting in the reduction of the expression of the PnTCP2 protein, and promoted the formation of leaf margin lobes. At the same time, the downregulation of *PnTCP2* and *PnLBD41* led to the inhibition of the formation of PnTCP2-PnLBD41, alleviated the inhibition of the *PnKNAT6* promoter, and increased the expression of *PnKNAT6*, which was conducive to the formation of leaf margin lobes.

## Figures and Tables

**Figure 1 ijms-23-13296-f001:**
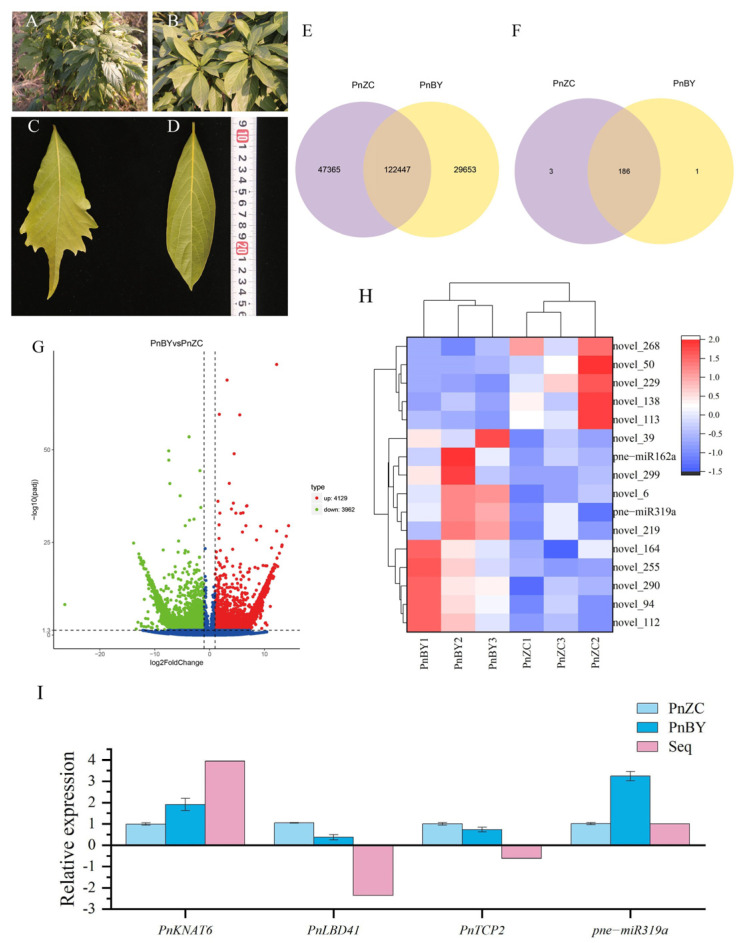
Leaf shape comparison and differential expression analysis between *P. neurantha* and *P. neurantha* var. *lobophylla.* (**A**) Branches of *P. neurantha* var. *lobophylla*. (**B**) Branches of *P. neurantha*. (**C**) A lobed leaf of *P. neurantha* var. *lobophylla*. (**D**) Leaf of *P. neurantha*. (**E**) Venn diagram showing the number of unigenes in PnBY and PnZC. (**F**) Venn diagram showing the number of miRNAs in PnBY and PnZC. (**G**) Volcanic map analysis of DEGs in PnBY. (**H**) Heat map analysis of differentially expressed miRNAs in PnBY. The unit of measurement on the tape is centimeters (cm). (**I**) Expression of leaf-lobed-related genes and miRNA quantified by transcriptome sequencing and qRT-PCR analysis. The standard deviation (SD) was used to measure the error within a group. The significance between samples was compared (*p* < 0.05).

**Figure 2 ijms-23-13296-f002:**
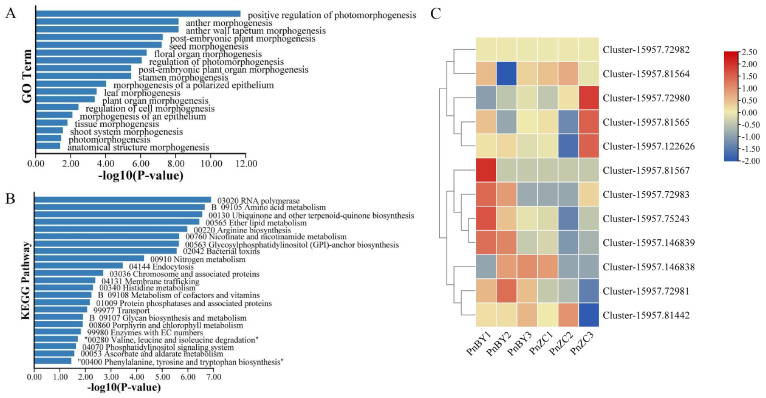
GO, KEGG enrichment analysis, and heat map analysis of target genes. (**A**) GO enrichment analysis of the target genes of differentially expressed miRNAs. Only GO subcategories related to morphogenesis are listed in the figure. (**B**) KEGG enrichment analysis of the target genes. (**C**) Heat map analysis of the target gene of pne-miRNA319a.

**Figure 3 ijms-23-13296-f003:**
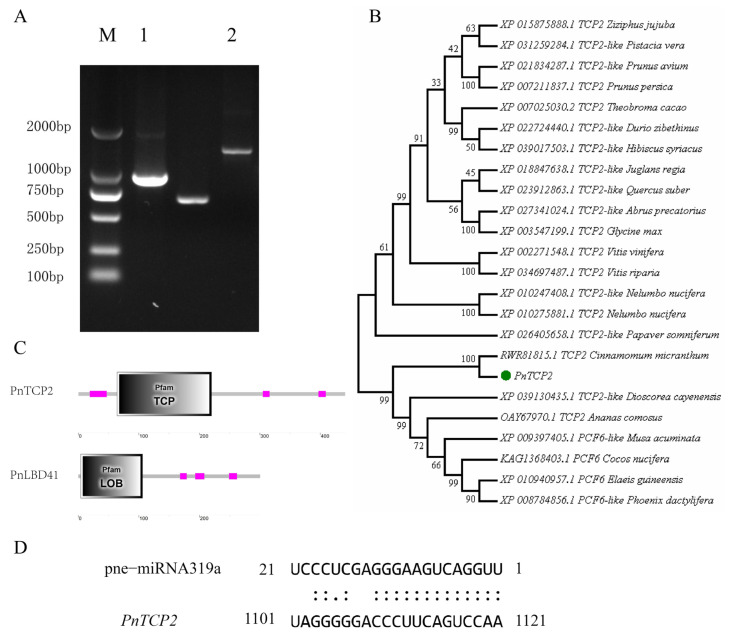
*PnTCP2* and *PnLBD41* sequence analyses. (**A**) *PnTCP2* and *PnLBD41* gel electrophoresis diagrams. (**B**) Phylogenetic tree of the PnTCP2 protein. (**C**) PnTCP2 and PnLBD41 protein knot conserved domain analysis (SMART). (**D**) Predicted target mRNA cutting-site scheme. Note: (**A**) M, 2000 bp DNA marker; 1, *PnLBD41* fragment; 2, *PnTCP2* fragment, (**C**) Purple square represents cooked coil.

**Figure 4 ijms-23-13296-f004:**
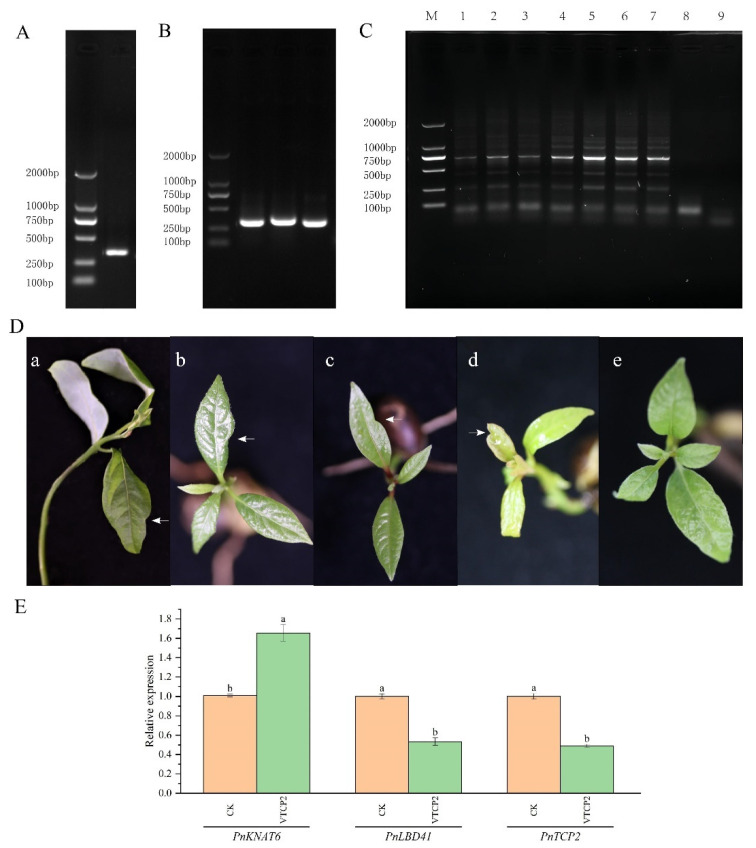
TVR-VIGS gene silencing function verification of *PnTCP2*. (**A**) Electrophoresis of *VTCP2* fragments. (**B**) PCR identification of positive clones. (**C**) Test electrophoresis diagram of TRV-VIGS-positive plants. (**D**) Lobed leaf phenotype after *PnTCP2* VIGS gene silencing. The position pointed out by the white arrow is the lobed part of the blade. (**E**) Expression analysis of related genes in pTRV2-*VTCP2* seedlings. The standard deviation (SD) was used to measure the error within a group, and the Tukey method was used to compare the significance between samples at *p* < 0.05. Note: (**C**) M, the 2000 pb DNA marker; (1, 2 and 3) control bands, (4–7) pTRV2-*VTCP2* plant bands, (8,9) bands of plants without TVR-VIGS treatment. (**a**–**d**) The pTRV2-*VTCP2* gene-silenced seedlings of *P. neurantha.* (**e**) The control treatment seedlings of *P. neurantha*. (**E**) Different lowercase letters indicate significant differences between the two groups (*p* < 0.05).

**Figure 5 ijms-23-13296-f005:**
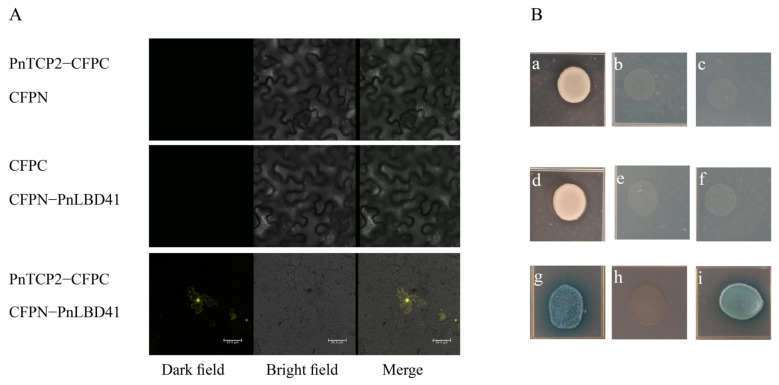
Verification of the protein interaction. (**A**) BiFC verification of the interaction between related proteins. (**B**) Yeast two-hybrid verification of the interaction between related proteins. Note: (**A**) PnTCP2 and PnLBD41 protein interactions. (**B**) (**a**) Yeast strain Y2H containing the pGBKT7-PnTCP2 plasmid grown in SD/–Trp medium, (**b**) Yeast strain Y2H containing the pGBKT7-PnTCP2 plasmid grown in SD/–Trp–Ade–His medium, (**c**) Yeast strain Y2H containing the pGBKT7-PnTCP2 plasmid grown in SD/–Trp–Leu–Ade–His medium, (**d**) Yeast strain Y187 containing the pGATD7-PnLBD41 plasmid grown in SD/–Leu medium, (**e**) Yeast strain Y187 containing the pGATD7-PnLBD41 plasmid grown in SD/–Lue–Ade–His medium, (**f**) Yeast strain Y187 containing the pGATD7-PnLBD41 plasmid grown in SD/–Trp–Leu–Ade–His medium, (**g**) Positive control, (**h**) Negative control, (**i**) PnTCP2 and PnLBD41 protein interaction. The size of the scale in the figure is 50.0 μm.

**Figure 6 ijms-23-13296-f006:**
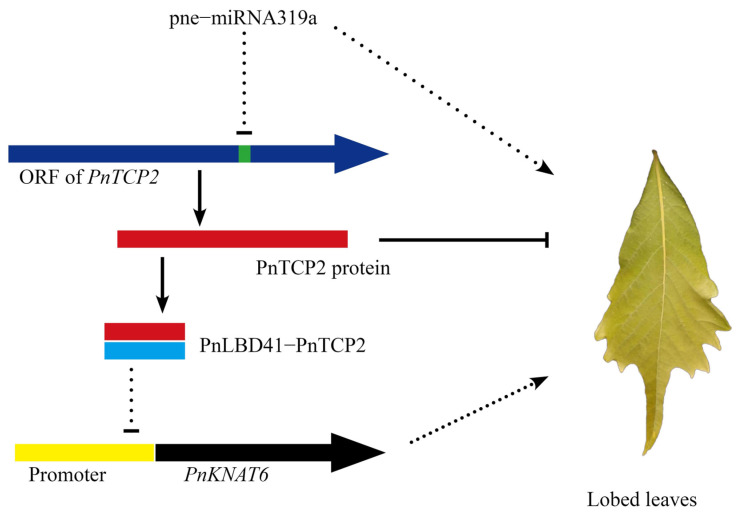
Theoretical schematic diagram of *PnTCP2* regulating the formation of leaf margin lobes in *P. neurantha* var. *lobophylla.* Arrows and arrowheads indicate regulatory effects. Lines with terminal bars indicate repressive effects. The regulatory relationship represented by dotted lines comes from previous reports and has not been verified in this study.

## Data Availability

The data presented in this study are available on request from the corresponding author.

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
