# Peer review of "The Role of PnTCP2 in the Lobed Leaf Formation of Phoebe neurantha var. lobophylla"

_ijms, 2022, doi:10.3390/ijms232113296_

Round 1

Reviewer 1 Report

Minor comment

The authors gave a detailed description of the process of filtering the source reads before assembling using Trinity. But I didn't find any indication in the text with which software the filtering was done. If it was made with the help of homemade scripts, it is worth specifying their name and putting them in open access.

Major comments:

After the assembly, the authors used clustering to get Unigenes. I have concerns that this could lead to the loss of information about transcripts of paralogous genes. It would be worth discussing this issue in the paper and showing that the clustering procedure does not lead to a strong collapse of paralogs. It is also necessary somewhere in the paper to provide information about the number of transcripts originally assembled by Trinity.

Author Response

Point 1: The authors gave a detailed description of the process of filtering the source reads before assembling using Trinity. But I didn't find any indication in the text with which software the filtering was done. If it was made with the help of homemade scripts, it is worth specifying their name and putting them in open access.

Response 1: Trimmatic v0.38 software was used to filter raw data,and It has been modified in "4.3. Analysis of sequencing data" of the manuscript.

Point 2: After the assembly, the authors used clustering to get Unigenes. I have concerns that this could lead to the loss of information about transcripts of paralogous genes. It would be worth discussing this issue in the paper and showing that the clustering procedure does not lead to a strong collapse of paralogs. It is also necessary somewhere in the paper to provide information about the number of transcripts originally assembled by Trinity.

Response 2: Please provide your response for Point 2. (in red)It has been modified in "2.1. Identification and analysis of DEGs" and "Discussion" of the manuscript. The details are as follows:

(in 2.1. Identification and analysis of DEGs)

The transcript sequence obtained by Trinity splicing is used as the reference sequence for subsequent analysis. The longest cluster sequence obtained from Corset hierarchical clustering is used as the Unigenes for subsequent analysis. The length of transcripts and Unigenes were counted respectively, and the results are shown in Figure 2. After Corset hierarchical clustering, the N50/N90 of the sequence increased from 1170/319 bp of the transcripts to 1451/497 bp of Unigenes, and the Mean length of the sequence also increased from 764 bp to 1043 bp. The results showed that Unigenes could better represent the transcripts for subsequent analysis. ............

Figure 2. Length distribution statistics of transcripts (A) and Unigenes (B).

(in 3. Discussion)

In this study, Corset hierarchical clustering method was used to select Unigenes, which not only achieved the goal of aggregating redundant transcripts, but also improved the detection rate of differentially expressed genes27-29.

27 Davidson, N. M. & Oshlack, A. Corset: enabling differential gene expression analysis for de novo assembled transcriptomes. BioMed Central 15, 410, doi:10.1186/s13059-014-0410-6 (2014).

28 Chen, L. Y., Morales-Briones, D. F., Passow, C. N. & Yang, Y. Performance of gene expression analyses using de novo assembled transcripts in polyploid species. Bioinformatics 35, 4314-4320, doi:10.1093/bioinformatics/btz620 (2019).

29 Hébert, F. o. O., Grambauer, S., Barber, I., Landry, C. R. & Aubin-Horth, N. Transcriptome sequences spanning key developmental states as a resource for the study of the cestode Schistocephalus solidus, a threespine stickleback parasite. Gigascience 5, 24, doi:10.1186/s13742-016-0128-3 (2016).

Reviewer 2 Report

Dear Sun and associates,

In the submitted manuscript, Sun and associates explored the role of PnTCP2 in lobe formation in Phoebe neurantha. To examine the role of PnTCP2 in lobe formation, PnTCP2 was silenced by VIGS. Additionally, the authors showed interaction of PnTCP2 with PnLBD41. Also, they performed RNA-seq and sRNA-seq analyses in two varieties of Phoebe neurantha (lobed and unlobed). Based on these analyses, they speculated about the role of pne-miR319a, PnLBD41and PnAS1 in lobe formation. They based their model for the miRNA319a-PnTCP2 module on known data from other plants, mostly Arabidopsis, although Magnoliids, dicots, and monocots diverged after emerging angiosperms (100-120 mya).

However, speculations are not provided a solid platform for role establishment for miRNA319a, PnLBD41 and PnAS1 in lobe formation. Consequently, additional VIGS experiments should be performed to prove their roles in lobe formation.

In my opinion, it is possible to publish the manuscript in IJSM after some changes in the manuscript and resubmission or to perform additional experiments (which option is preferable, I leave the decision to the authors).

Best regards

Major comments

1.     English language: Generally, it looks good, but please edit it again.

For example, the “module” is suited more than the “model” in this case. See the difference between module and model.

Are you sure that the name of the unlobed, unserrated, smooth leaf is a whole leaf?

It is the entire leaf, isn’t it?

Line 157 PnLBD41 protein interacted: interacts. It continues to happen in the plant.

2.     Introduction:

a.     Remove unimportant and unscientific descriptions from the first paragraph (royal family, ancient China, beautiful, etc.)-it is not a point of your manuscript.

b.     Enlarge the description of known models of lobe formation. Please provide additional examples and references for genes and miRs in plants from families other than Brassica. I expected that you would cite the works from Sinha N, Arazi T, Ori N, and other labs that used plants from other families to explore lob formation.

Apparently, mutations in Arabidopsis do not predict the same effect in other plants. It was shown a lot of times.

3.     Results and discussion:

a.     Only knock-out or silencing mutants can reveal the gene roles. You used VIGS only for TCP2. Consequently, you can talk about the role of TCP2 only. All other, it is speculation.

You have two options:

1. It is fine to speculate in the discussion. Remove miR319 from the article name (including module/model), abstract, and Fig 5 and/or write in Fig 5 that it is the prediction for miR, LBD41 based on other works. The difference in the VIGS experiment is less than two, and lobes are very tiny compared to the lobed variety. So, in the discussion, please discuss additional candidates for lobe phenotype.

2. You can perform additional VIGS experiments for miR319, PnLBD4, PnAS1.

b.     Also, in agro-infiltration experiments, you can show the cleavage of PnLTCP2 by pne-miR319a. Prediction is not always correct. However, I think that TCP2-miR319a is conservative cleavage. Write in Fig 3D that this is a prediction.

c.     Why did you call for miR319a from your plant nta-miR319a? It is a mistake. The name should be pne-miR319a for mature miR. Also, it is like for the gene and protein: gene encoded for miR is in italic; mature miR in roman text. Fix it through the manuscript.

d.     I did not find that you mentioned what you used in the statistical analysis of gene expression in qPCR: SE or SD. Write it. Also, if you used SE, please use SD. Also, write level of p.

e.     Table S2: give correct names for your miRs—for example, pne-miR319a. If you want to mention that it is novel- do a column for this. Also, enlarge the column width. Like in c: why do you use nta?

f.      For qPCR primers, add efficiency in a separate column.

Minor comments

1.     Please remove the word gene after PnTCP2: it is in italic, and also, you mentioned it earlier (“differentially expressed target genes”). Do it through the manuscript.

2.     Line 140 2.6 change identification on analysis.

3.     In all excel tables, enlarge the column width.

Author Response

Response to Reviewer 2 Comments

Point 1: English language: Generally, it looks good, but please edit it again.

For example, the “module” is suited more than the “model” in this case. See the difference between

module and model.

Are you sure that the name of the unlobed, unserrated, smooth leaf is a whole leaf?

It is the entire leaf, isn’t it?

Line 157 PnLBD41 protein interacted: interacts. It continues to happen in the plant.

Response 1: We made revisions in the manuscript. 

Point 2: Introduction:

  1. Remove unimportant and unscientific descriptions from the first paragraph (royal family,

ancient China, beautiful, etc.)-it is not a point of your manuscript.

  1. Enlarge the description of known models of lobe formation. Please provide additional

examples and references for genes and miRs in plants from families other than Brassica. I expected

that you would cite the works from Sinha N, Arazi T, Ori N, and other labs that used plants from

other families to explore lob formation.

Apparently, mutations in Arabidopsis do not predict the same effect in other plants. It was shown a

lot of times.

Response 2: 

  1. IntroductionPhoebe neurantha is an evergreen tree of the Lauraceae family. Phoebe neurantha (Hemsl.) Gamble var. lobophylla Y. J. Fei (Figure 1A) is a new variety with a different leaf morphology,.....”

We deleted the unimportant and unscientific descriptions in the manuscript as above.   

  1.  We added some research reports in other plants. The details are as follows:

“Researchers have identified many genes related to leaf margin morphology and development in different plants, such as AtCUC22 and AtTCP43 in Arabidopsis, CcLBD414 in Celosia cristata, MtLMI15 in Medicago truncatula, and KNOX1, which is more widely reported (in addition to reports in Arabidopsis6, there are also reports in Solanum lycopersicum7, FernMickelia scandens8, Pisum sativum9, Medicago truncatula5,7 and other plants).”

2 Zheng, G. et al. Conserved and novel roles of miR164-CUC2 regulatory module in specifying leaf and floral organ morphology in strawberry. New Phytologist 224, 480-492, doi:10.1111/nph.15982 (2019).

3 Challa, K. R., Rath, M. & Nath, U. The CIN-TCP transcription factors promote commitment to differentiation in Arabidopsis leaf pavement cells via both auxin-dependent and independent pathways. Plos Genetics 15, doi:10.1371/journal.pgen.1007988 (2019).

4 Meng, L.-S. et al. Modification of flowers and leaves in Cockscomb (Celosia cristata) ectopically expressing Arabidopsis ASYMMERTIC LEAVES2-LIKE38 (ASL38/LBD41) gene. Acta Physiologiae Plantarum 32, 315-324, doi:10.1007/s11738-009-0409-x (2010).

5 Wang, X. et al. LATE MERISTEM IDENTITY1 regulates leaf margin development via the auxin transporter gene SMOOTH LEAF MARGIN1. Plant Physiology (2021).

6 Rast-Somssich, M. I. et al. Alternate wiring of a KNOXI genetic network underlies differences in leaf development of A. thaliana and C. hirsuta (vol 29, pg 2391, 2015). Genes & Development 30, 132-132, doi:10.1101/gad.276121.115 (2016).

7 Hokuto, N. et al. Leaf form diversification in an ornamental heirloom tomato results from alterations in two different HOMEOBOX genes. Current biology: CB 31, 4788-4799.e4785, doi:10.1016/J.CUB.2021.08.023 (2021)

8 Cruz, R., Melo-de-Pinna, G. F. A., Vasco, A., Prado, J. & Ambrose, B. A. Class I KNOXIs Related to Determinacy during the Leaf Development of the FernMickelia scandens (Dryopteridaceae). International Journal of Molecular Sciences 21, doi:10.3390/ijms21124295 (2020).

9 Moreau, C., Hofer, J. M. I., Eléouët, M., Sinjushin, A. & Ellis, T. H. N. Identification of Stipules reduced, a leaf morphology gene in pea (Pisum sativum ). New Phytologist 220, 288-299, doi:10.1111/nph.15286 (2018).

Point 3:

Results and discussion:

  1. Only knock-out or silencing mutants can reveal the gene roles. You used VIGS only for TCP2.

Consequently, you can talk about the role of TCP2 only. All other, it is speculation.

You have two options:

  1. It is fine to speculate in the discussion. Remove miR319 from the article name (including

module/model), abstract, and Fig 5 and/or write in Fig 5 that it is the prediction for miR, LBD41 based

on other works. The difference in the VIGS experiment is less than two, and lobes are very tiny

compared to the lobed variety. So, in the discussion, please discuss additional candidates for lobe

phenotype.

  1. You can perform additional VIGS experiments for miR319, PnLBD4, PnAS1.
  2. Also, in agro-infiltration experiments, you can show the cleavage of PnLTCP2 by pne

miR319a. Prediction is not always correct. However, I think that TCP2-miR319a is conservative

cleavage. Write in Fig 3D that this is a prediction.

  1. Why did you call for miR319a from your plant nta-miR319a? It is a mistake. The name should

be pne-miR319a for mature miR. Also, it is like for the gene and protein: gene encoded for miR is in

italic; mature miR in roman text. Fix it through the manuscript.

  1. I did not find that you mentioned what you used in the statistical analysis of gene expression

in qPCR: SE or SD. Write it. Also, if you used SE, please use SD. Also, write level of p.

  1. Table S2: give correct names for your miRs—for example, pne-miR319a. If you want to

mention that it is novel- do a column for this. Also, enlarge the column width. Like in c: why do you

use nta?

  1. For qPCR primers, add efficiency in a separate column.

Response 3: 

We have modified the description of the title, abstract and related pictures, and identified the conclusions inferred from previous research results. As for the question "lakes are very tiny compared to the lobed variety", we made the following reply during the discussion:

“VIGS technology has the advantages of independent genetic transformation, convenient inoculation and high silencing efficiency, which can achieve silencing of targeted genes in contemporary plants36. However, there are still some limitations, such as being unable to completely silence genes37. In this study, we successfully silenced PnTCP2 in the leaves of Phoebe neurantha seedlings through VIGS technology, and obtained the phenotype of lobed leavse. However, the lobes of gene silenced leaves are very tiny compared to the lobed leavse of Phoebe neurantha var. Lobophylla. The reason for this phenomenon may be that the mRNA of PnTCP2 was not completely silenced, and the surviving mRNA weakens the degree of phenotype. Moreover, the regulation of plant leaf margin morphology involves the synergistic effect of multiple genes (including TCP43, CUC238, LMI111, etc.). Genes such as PnTCP4 were not silenced. The expression of these genes weakens the phenotypic changes caused by PnTCP2 silencing. However, the fact that the silence of PnTCP2 caused the leaves of Phoebe neurantha seedlings to be lobed cannot be concealed. Therefore, PnTCP2 and the possible pne-miRNA319a-PnTCP2 module played an important regulatory role in lobed leaf formation of Phoebe neurantha var. Lobophylla.

b.

The corresponding position of the manuscript has been marked: ". (D) Predicted target mRNA cutting site scheme".

c.

All "nta -" in the manuscript (including tables and pictures) have been changed to "pne -".

d.

The description of qPCR related pictures has been modified as follows: "The standard deviation (SD) was used to measure the error within the group, and the Tukey method was used to compare the significance between samples at p=0.05." has been added.

e.

The table has been modified and adjusted according to the modification comments.

f.

It has been added in the form according to the modification comments.

Point 4:

Minor comments

  1. Please remove the word gene after PnTCP2: it is in italic, and also, you mentioned it earlier

(“differentially expressed target genes”). Do it through the manuscript.

  1. Line 140 2.6 change identification on analysis.
  2. In all excel tables, enlarge the column width.

Response 3: 

It has been revised and adjusted in the manuscript and table according to the revision comments

Round 2

Reviewer 1 Report

No additional comments.

Author Response

Thank you for your suggestions on revising my manuscript. These suggestions are very pertinent and helpful to improve the quality of my manuscript. Thanks again.

Reviewer 2 Report

Dear Sun and associates,

In the submitted manuscript, Sun and associates explored the role of PnTCP2 in lobe formation in Phoebe neurantha.

You fixed the problematic scientific points.

However, it would be best if you revised the English language again.

Additionally, you are continuing to use some terms not in the correct form, despite being asked to change them in the first round. So, I choose the option "major revision" because of this. The mature miR you write in roman. Gene encoded for miR in italic. In most cases, it should be mature miR.

Please look comments.

I think it is possible to publish the manuscript in IJSM after an additional revision round.

 Best regards

Comments

All line numbers are related to the second version of the manuscript without track changes.

1.     Response 1: Response 1: We made revisions in the manuscript.

It was not enough, and you added new mistakes. I am sure that I did not list all mistakes. So, please revise it again. 

 English language: still should be fixed:

Line 27: played = plays

Line 39:  trew? What do you mean Sassafras Trew?

Line 48: FernMickelia scandens: fern in lowercase all letters, add space after fern. Mickelia scandens in italic. Fix this reference also.

Line 166: were-was. down regulated without the space and hyphen (downregulated).

Line 168: upregulated without the space and hyphen

Line 191: Different plants have great differences-please change one of the words by synonym

Line 193: you do not need "many"

Lines 238, 239: leavse-leaves

Line  246: play

 2.     New Figure 2-it is better to move it to supplement. If you decide to leave it in the main text, change it to figure 1.

3.     c.

All "nta -" in the manuscript (including tables and pictures) have been changed to "pne -".

Why you did not write mature miRs in roman and leave them in italic?

Line 26, 30: pne- miRNA319a-change on roman text, remove the space between pne to miR. miRNA319a-roman also.

For mature miR you do not use italic!!!. Italic only for genes encoded for miRs.

Lines 61-63, 88-90, 107, 119, 121, 124-126, 132, 143, 145, 147, 210, 211, 212, 214, 222, 217, 250, 248, 251, 246: all miRs should be in roman.

4.     Line 90: pnapne-?? Fix it.

5.     Lines 99-100. The standard deviation (SD)-I found it in Figure 5. Add it to fig 1 I.

p should be written p < 0.05.

6.     Line 104: related in lowercase

7.     Line 136: ofPnTCP2-add space after of. I saw it in several places. I hope, that I will mention them all. Please check again all text.

8.     Lines 140, 141: remove pfam

9.     Line 161: (Tukey, p < 0.05)-transfer after down regulated. Downregulated without word space. (Figure 5E) to the end of the sentence.

10.  Lines 169-171-Remove the word protein after protein names

11.  Line 172: (Figure 5G)- to the end of the sentence. I think it is better to write that YTH supported BiFC, and not validation.

12.  Line 181: p=0.05. the same mistake that was mentioned before. Interesting-in text you wrote as needed , p < 0.05)

13.  213-, 220: reference numbers with space from the words, remove the spaces

14.  Line 243: remove etc.

Author Response

(The authors gave the same response as above.)

Round 3

Reviewer 2 Report

Dear Sun and associates,

I think it is possible to publish the manuscript in IJSM.

I have only minor comments:

Line 28 (in the final version with track changes): I think that it is better to leave important.

Line 40: Phoebe Nees are entire leaves=Phoebe Nees has

Line 43: remove therefore

Line 152: remove n, leave a 

Line 184: support, remove ed

Line 202: disperancies does not suit here, try to remove first word different. Plants have great differences in leaf morphology

 Best regards

Author Response

Dear Sir or Madam

Thank you very much for your suggestions on the revision of my manuscript, which has been revised according to your suggestions in the latest version of the manuscript. See the attachment for details. Thank you very much.
